# A Green, Rapid and Efficient Dual-Sensors for Highly Selective and Sensitive Detection of Cation (Hg^2+^) and Anion (S^2−^) Ions Based on CMS/AgNPs Composites

**DOI:** 10.3390/polym12010113

**Published:** 2020-01-05

**Authors:** Yun Xue, Lina Ma, Lei Zhang, Wanting Zhao, Zichao Li, Qun Li

**Affiliations:** 1College of Chemistry and Chemical Engineering, Qingdao University, Qingdao 266071, China; xueyun@qdu.edu.cn (Y.X.); malina@qdu.edu.cn (L.M.); 2017020832@qdu.edu.cn (L.Z.); 2016020406@qdu.edu.cn (W.Z.); 2College of Life Sciences, Qingdao University, Qingdao 266071, China; zichaoli@qdu.edu.cn

**Keywords:** nano-silver, optical detection, sensitivity and selectivity, mercury ions, sulfide ions

## Abstract

Detection of mercury (Hg^2+^) and sulfide (S^2−^), universal and well-known toxic ions, is crucial in monitoring several diseases. How to design and fabricate the high-performance sensor for simultaneously and accurately detecting the Hg^2+^ and S^2−^ is critical. Herein, we proposed a novel and convenient strategy for optical detection of Hg^2+^ and S^2−^ by employing a carboxymethyl cellulose sodium/silver nanoparticle (CMS/AgNPs) colloidal solution, in which AgNPs were used as monitor for Hg^2+^ and S^2−^, and the CMS was utilized as both the stabilizer and the hydrophilic substrate for AgNPs. Well-identifiable peaks for Hg^2+^ and S^2–^ were obtained in water based on UV–VIS absorption spectra, the absorbance intensity and/or position of nano-silver vary with the addition of Hg^2+^ cation and S^2–^ anion, accompanying with color change. Impressively, the optimal AgNPs anchored CMS exhibited a high sensitivity and selectivity toward Hg^2+^ and S^2−^, the change in absorbance was linear with the concentration of Hg^2+^ (0–50 μM) and S^2−^ (15–70 μM), and the lowest limits of detection (LOD) were 1.8 × 10^−8^ M and 2.4 × 10^−7^ M, respectively. More importantly, owing to the superior properties in testing Hg^2+^ and S^2−^, the fabricated sensor was successfully applied for detection of target ions in lake and tap water samples. All these good results implied that the designed strategy and as-designed samples is promising in detecting cation (Hg^2+^) and anion (S^2−^) ions and open up new opportunities for selecting other kinds of ions.

## 1. Introduction

Nowadays, water pollution has been considered as one of the most serious and dangerous forms of pollution, which has aroused numerous environmental problems and drawn growing attention [1,2,3,4]. Mercury (Hg^2+^) and sulfide (S^2−^) are well-known toxic ions and hazardous chemicals. Hg^2+^ exists widely in air, soil, and water [5,6]. It is very detrimental to human health, even at low concentrations, causing serious diseases to the heart, liver, kidneys, endocrine, brain, and nervous system [7,8]. S^2−^ not only exists as a hazard to living systems but can also easily convert into H_2_S in acidic media [9], and H_2_S is harmful to our eyes, causing conjunctivitis and other related eye diseases [10]. At certain concentrations, it will bind to hemoglobin in the blood and cause poisoning and even death [11]. Thus, the development of efficient sensors for the detection of S^2−^ and Hg^2+^ is of considerable urgency.

Up to now, sophisticated instruments, such as cold vapor atomic absorption spectrometers (CVAAS), atomic fluorescence spectrometers (AFS), inductively coupled plasma mass spectrometers (ICP-MS), stripping voltammeters, and inductively coupled plasma optical emission spectrometers (ICP-OES) have been used for the analysis of Hg^2+^ [12,13]. Infrared absorption spectroscopy, ICP-OES, ultraviolet fluorescence method (GC-TS), and X-ray fluorescence spectrometry are used for analyzing S^2−^ [14]. However, these methods always need a long analysis time, precision equipment, complicated procedures, some technical professionals, or significant special skill. Therefore, it is very necessary to develop some simple, routinely-monitored selective and sensitive techniques for Hg^2+^ and S^2−^ in environmental water samples.

In recent years, nanoparticle-based colorimetric probes have received significant attention due to their high selectivity and sensitivity [15,16,17,18,19]. Various functionalized Ag and Au nanoparticles have been extensively developed for the selective and sensitive detection of S^2−^ and Hg^2+^ [20,21,22,23,24]. Nevertheless, such strategies also suffer from some drawbacks, including complicated synthesis, high detection limit, and single ion detection [25,26,27,28]. UV–VIS spectra, in stark contrast, have been verified to be a more-powerful optical technique for the exploitation of rapid, efficient, and low concentration analyses due to their reliable, low cost, and rapid implementation. From the viewpoint of practical application, an excellent sensor should not only have highly sensitive and selective performance, an environmentally friendly synthetic route, and have a simple and economical operation, but also show rapid and efficient detection toward multiple ions simultaneously. To fulfill the above requirements, we propose a novel CMS/AgNPs colloidal optical sensor to analyze Hg^2+^ and S^2−^ in aqueous samples. Notably, our method has several advantages: Firstly, all material reagents are green, non-toxic, and low-cost. Secondly, the synthesis and detection reactions are simple, environmental-friendly, and rapid. Thirdly, the prepared material can act directly as a sensor, which is highly sensitive and has selective detection for multiple ions, cations (Hg^2+^) and anions (S^2−^). Fourthly, the LOD is low and the color changes are obvious and visible by the naked eye. Furthermore, the system can be utilized directly in lake and tap water without any other treatment.

## 2. Experimental

### 2.1. Reagents and Apparatus

AgNO_3_, NH_3_·H_2_O, ascorbic acid (Vc), carboxymethyl cellulose sodium (CMS), Al(NO_3_)_3_·9H_2_O, Cu(NO_3_)_2_·3H_2_O, Bi(NO_3_)_3_·5H_2_O, Ba(NO_3_)_2_, Ca(NO_3_)_2_·4H_2_O, Cd(NO_3_)_2_·4H_2_O, Co(NO_3_)_2_·6H_2_O, Fe(NO_3_)_2_·6H_2_O, Fe(NO_3_)_3_·9H_2_O, KNO_3_, Mg(NO_3_)_2_·6H_2_O, Ni(NO_3_)_2_·6H_2_O, Pb(NO_3_)_2_, Zn(NO_3_)_2_·6H_2_O, HgCl_2_, NaNO_3_, NaNO_2_, Na_2_CO_3_, NaHCO_3_, Na_2_HPO_4_, NaH_2_PO_4_, Na_3_PO_4_, NaF, NaCl, NaBr, NaI, Na_2_S, Na_2_SO_3_, Na_2_S_2_O_8_, and Na_2_SO_4_ are of analytical reagent grade and were purchased from Sinopharm Chemical Reagent Co., LN (Shanghai, China). The concentration of salt solution is 1 × 10^−2^ M. All glassware used in the experiment were cleaned well using HNO_3_ and rinsed with distilled water, followed by oven drying at 100 °C for use.

### 2.2. Synthesis of CMS/AgNPs Colloidal Solution

The AgNPs aqueous dispersion was synthesized by the chemical reduction of Tollens’ reagent using Vc as an environmentally benign reducing agent, and CMS as a non-toxic stabilizing agent, respectively. A total of 100 μL of 0.1 M Tollens’ reagent was added to CMS aqueous solution (0.1% (*w*/*v*), 100 mL) under constant stirring. The mixture was continued to be stirred for 10 min to allow the diffusion of metal ions into the stabilizing agent. After that, Vc (0.1 M, 100 μL) was added under continuous stirring for 20 min. The entire process was carried out at a constant temperature of 70 °C.

### 2.3. Characterization of CMS/AgNPs Colloidal Solution

The UV–VIS absorption spectra were recorded via UV–VIS spectrophotometer (UV-2450, Shimadzu, Kyoto, Japan) with a variable wavelength between 300 and 600 nm at room temperature. X-ray diffractometer (XRD) measurements was carried out on a powder X-ray diffractometer (D/MAX-RB, Tokyo, Japan) using CuKα radiation (λ = 0.15418 nm) over the 2θ range of 5°–80° with a step size of 0.05°. Transmission electron microscopy (TEM) images were obtained by a JEOL JEM-2100 Plus microscope (JEOL, Tokyo, Japan) at an accelerating voltage of 200.0 kV, the samples were prepared by dropping the solution containing CMS/AgNPs colloidal solution onto the carbon-coated copper grid and dried in vacuum.

### 2.4. Colorimetric Detection of Hg^2+^ and S^2−^

The CMS/AgNPs colloidal solution was diluted three times using ultra-pure water for the colorimetric determination of Hg^2+^. To investigate the sensitivity and selectivity of CMS/AgNPs colloidal solution; Firstly, 15 kinds of metal ions (15 µL, 10^−2^ M) including Mg^2+^, Co^2+^, Cu^2+^, Ni^2+^, Zn^2+^, Ca^2+^, Cd^2+^, Fe^3+^, Fe^2+^, K^+^, Ba^2+^, Na^+^, Hg^2+^, Al^3+^, and Pb^2+^ ions were respectively added into the CMS/AgNPs colloidal solution (3 mL). Secondly, in order to obtain the detection limit, Hg^2+^ ions of different concentrations (0.3–360 µL, 1 mM) were added into the CMS/AgNPs colloidal solution (3 mL). Finally, the absorption spectra of CMS/AgNPs with various metal ions (50 µM) were obtained by a UV–VIS spectrophotometer and color changes were observed by the naked eye. The Hg^2+^ or Cu^2+^ ions (50 µM) were added to CMS/AgNPs colloidal solution (3 mL) for TEM test.

In order to detect S^2−^, the CMS/AgNPs colloidal solution was diluted twice with ultra-pure water. The detection method is similar with that of Hg^2+^, 15 kinds of negative ions (30 µL, 10^−2^ M), including NO_3_^−^, NO_2_^−^, CO_3_^2−^, HCO^3−^, HPO_4_^2−^, H_2_PO_4_^−^, PO_4_^3−^, F^−^, Cl^−^, Br^−^, I^−^, SO_3_^2−^, S_2_O_8_^2−^, S^2−^, and SO_4_^2−^, were added into the CMS/AgNPs colloidal solution (3 mL), respectively. Then, different concentrations of S^2−^ ions (3–210 µL, 1 mM) were added to the CMS/AgNPs colloidal solution (3 mL). Finally, color changes were observed by the naked eye and the absorption spectra of the above solutions were obtained by UV–VIS spectrophotometer. The TEM was obtained by CMS/AgNPs colloidal solution containing S^2−^ or SO_4_^2−^ ions with a concentration of 50 µM. The LOD is the lowest concentration of the measured sample, using the formula [16,20]: LOD = 3 S/N (S: the standard deviation of the blank sample, N: the slope of the standard linear).

## 3. Results and Discussion

### 3.1. Preparation of CMS/AgNPs Colloidal Solution

When mixed with water, the CMS will form a gel because of the abundance of carboxyl groups on the surface of the CMS. The negative charge, such as that caused by carboxyl groups, can attract a positive charge [Ag(NH_3_)_2_]^+^ toward its polymer chains. Then the [Ag(NH_3_)_2_]^+^ around the carboxyl group was reduced in the presence of the reducing agent Vc and nucleated silver nanoparticles (Figure 1). The concentration of nano-silver is about 10^−4^ M in CMS/AgNPs colloidal solution. Furthermore, the UV–VIS spectral responses of the CMS/AgNPs were recorded at different pH in Appendix A. The pH of the synthesized CMS/AgNPs sol is 8.39. The CMS/AgNPs shows no noticeable effects with the change in pH from 6.59 to 10.51. However, the CMS/AgNPs shows spectral changes at low pH (<2.48) due to the COO^−^ in the CMS combined with H^+^ to form COOH, which reduced the stabilizing of AgNPs. These results confirm that CMS can stabilize AgNPs because of electrostatic adsorption between COO^−^ and [Ag(NH_3_)_2_]^+^.

Figure 2 is the optimal screening of the experiment. The absorbance of AgNPs increases with the increase of the amount of silver ions in Figure 2A. The absorbance of AgNPs decreases and red shift occurs in 200 μL of Tollens’ reagent, indicating that the nano-silver particles become larger at a higher concentration, so the optimal amount is 100 μL of Tollens’ reagent. The absorbance of AgNPs increases with the increase in the amount of Vc in Figure 2B. The absorbance of AgNPs decreases and red shift occurs in 200 μL Tollens’ reagent, indicating that the nano-silver particles become larger at a higher concentration, so the optimal amount is 100 μL of Tollens’ reagent. Figure 2C shows the UV–VIS spectra of CMS/AgNPs by synthesis with different reducing agents, indicating that Vc is weaker in reducing ability than NaBH4, but stronger than glucose and sodium citrate. The latter two require a catalyst for the reaction to synthesize AgNPs. The synthesized AgNPs with NaBH_4_ was weaker in stability than Vc, so Vc was the best, green reducing agent. The stability of the synthesized AgNPs is optimal in Figure 2D. We can see from Figure 2 that the best experimental conditions are CMS aqueous solution (0.1% (*w*/*v*), 100 mL) and 1:1 molar ratio of Tollens’ reagent (0.1 M, 100 μL) to Vc (0.1 M, 100 μL).

### 3.2. Characterization Studies of AgNPs

UV–VIS spectra demonstrated that AgNPs displayed a strong characteristic absorption peak within the visible region from 410 to 430 nm by comparing to CMS. The maximum absorption peak was located at 420 nm (Figure 3A) and the CMS/AgNPs colloidal solution was transparent yellow (inset, Figure 3A). The XRD pattern of AgNPs is represented in Figure 3B. The 2θ diffraction peaks at 38.10°, 44.26°, 64.61°, and 77.46° were assigned to the (111), (200), (220), and (311) planes of crystallized silver with face-centered cubic (fcc) AgNPs, which suggest that silver nanoparticles were prepared. Figure 3C shows the FT-IR spectra of CMS and CMS/AgNPs. For CMS/AgNPs, the band centered at around 3456 cm^−1^ can be attributed to the stretching vibration of the hydroxyl group; the band at about 2885 cm^−1^ is assigned to the C–H group; the band at around 1605 cm^−1^ is attributed to the stretching vibration of the carboxyl group; the band at around 1396 cm^−1^ corresponds to the C–H bending mode; the absorption band at 1079 cm^−1^ is ascribed to C–O–C stretching mode from the glucosidic units; the peak at 612 cm^−1^ is related to the deformation vibration of hydrogen bond. Figure 3C shows the FT-IR spectra of CMS which is similar to the CMS/AgNPs, indicating that there is no chemical reaction between the formed AgNPs and the CMS.

TEM image showed that the spherical AgNPs crystals were well dispersed into CMS 3D nano networks (Figure 4A). The diameters of the AgNPs were in the range of 10–30 nm and the average particle size was 20 nm (Figure 4B). The selected area electron diffraction (SAED) image of CMS/AgNPs with bright circular rings confirmed the polycrystalline nature of the synthesized AgNPs (inset, Figure 4A). As shown in Figure 4C, energy dispersive spectroscopy (EDS) was further implemented to analyze the exist of Ag (3.0 KeV).

### 3.3. Mechanism for CMS/AgNPs Composites and the Detection of Hg^2+^ and S^2−^

CMS/AgNPs colloidal solution was prepared using ascorbic acid as the beginning reducing agent and Tollens’ reagent as the silver provider. The CMS was utilized as the stabilizer and reducing agent. Vc is a water-soluble vitamin and non-toxic reducing agent. There are double O–H with C=C bonds in the structure of Vc, so it has strong reducing ability. The synthesis process was carried out in an aqueous solution, and the synthesis method used a chemical in situ reduction method in Figure 5A. The synthesized raw materials were very cheap and green. Furthermore, the CMS/AgNPs colloidal solution could detect Hg^2+^ and S^2−^ ions as the colorimetric dual-sensors. As the concentration of mercury ions increased, it was observed that the color of CMS/AgNPs colloidal solution gradually changed from yellow to colorless by the naked eye, and the morphology also changed, as shown in Figure 8B. The size of AgNPs was changed from 10–30 nm to 100–200 nm in Figure 8C and induced the aggregation of AgNPs. The reason is that Ag and Hg^2+^ ions on the surface of AgNPs undergo a redox reaction to form an Ag–Hg amalgam [29]. The mechanism that Ag^0^ and Hg^2+^ undergo a redox reaction to form Ag^+^ and Hg^0^ on the surface of AgNPs is shown in Figure 5B. Since the generated Hg adheres to the surface of AgNPs, AgNPs become larger and agglomerate. The addition of S^2−^ ions results in a gradual color change of CMS/AgNPs from clear yellow to brown and even colorless, and the morphology of the CMS/AgNPs also changed from the original spheres to = amorphous particles. From TEM images (Figure 11B), we could see that the addition of S2− ions induced the aggregation of AgNPs and the size of AgNPs was changed from 10–30 nm to 40–80 nm, as shown in Figure 11C. The reason is that the addition of S2− induced the formation of Ag2S between Ag and S, as shown in Figure 5C and the aggregation of AgNPs [22]. The possible mechanism is as follows: Na_2_S is highly alkaline salt, S^2−^ ions is hydrolyzed in aqueous solution to form HS^−^, Ag^0^ is oxidized to Ag^+^ in the presence of O_2_ and HS^−^, then Ag^+^ and S^2−^ form Ag_2_S. However, the aggregation of AgNPs did not occur in the presence of any other ions, such as cations (Cu^2+^) and anions (SO_4_^2−^). This shows that the selectivity of CMS/AgNPs colloidal solution for Hg^2+^ and S^2−^ is very good.

### 3.4. Detection Selectivity and Sensitivity of Hg^2+^

Under optimum conditions, the sensitivity of CMS/AgNPs colloidal solution for Hg^2+^ was investigated. As observed in Figure 6A, the color of the sensing system gradually faded away with increasing amount of Hg^2+^ and the color changed to colorless when the concentration of Hg^2+^ was 50 μM, thus providing a platform for a colorimetric detection of Hg^2+^ by naked eyes. The absorption peak of AgNPs decreased gradually following the addition of Hg^2+^ and the maximum absorption wavelength of AgNPs displayed a distinct blue shift (Figure 6B). The reason is that Ag and Hg^2+^ ions on the surface of AgNPs undergo a redox reaction to form a Ag-Hg amalgam, which caused the reduction of the nano-silver concentration and smaller AgNPs, resulting in a decrease in the absorbance and a blue shift in the wavelength [29]. In order to analyze the correlation between ΔA and the Hg^2+^ concentration, the absorption changes with the addition of the ion concentration in the range of 0–120 µΜ were conducted. Based on these data, as shown in Figure 6C, there is a good linear relationship (R = 0.995) between ΔA and the Hg^2+^ (0–50 μM) concentration, and the LOD for Hg^2+^ was estimated to be 18 nM. Meanwhile, from Figure 6D, we can see a good linear relationship (R = 0.982) between Δl and the Hg^2+^ concentration in the range of 0-50 μM and the LOD was approximately 3.3 nM. From Appendix A, we can see that the stronger the reducing ability of the reducing agent, the smaller the average particle size of the synthesized AgNPs. Appendix A shows that the smaller the particle size of AgNPs, the lower the LOD of Hg^2+^. This phenomenon could be attributed to the high specific surface areas and surface energy of smaller size of AgNPs, when Hg^2+^ ions were added to the CMS/AgNPs colloid solution, the contact area and reactivity between AgNP and Hg^2+^ increases along with the size decrease of AgNPs, thus quickening the adsorption and reaction speed. Even at a low concentration of Hg^2+^, obvious absorbance signals can be obtained in the CMS/AgNPs colloid solution. These results indicated that the small nanoparticles have good sensitivity and lower LOD.

To evaluate the selectivity of CMS/AgNPs colloidal solution, besides Hg^2+^, 14 other kinds of metal ions, including Cd^2+^, Co^2+^, Cu^2+^, Fe^3+^, Fe^2+^, K^+^, Al^3+^, Ba^2+^, Ca^2+^, Mg^2+^, Na^+^, Ni^2+^, Pb^2+^, and Zn^2+^, were selected. The absorption spectra of AgNPs with various metal ions are given in Figure 7A. Comparing these plots, only Hg^2+^ had a response to AgNPs, and upon the addition of Hg^2+^, the absorption peak decreased and exhibited an obvious blue shift. For the 14 other metal ions, the absorption spectra were almost unchanged even at the maximum concentration (50 μM). As shown in Figure 7B, the change of ΔA was 0.36 in the presence of Hg^2+^, but there was no obvious change observed in the presence of other metal ions. The color of the sensing system did not change after the addition of other metal ions (50 μM) except for Hg^2+^, which made the system change from yellow to colorless (inset, Figure 7B). In addition, in the presence of all anions, the relative absorbance values were analogous to that of the Hg^2+^ value. These results clearly indicated that other related anions did not interfere with the spectral and colorimetric detection of Hg^2+^. The experimental results verify that our proposal shows unique selectivity for Hg^2+^.

Transmission electron microscopy (TEM) was used to study the nanoparticles’ size and the morphology of AgNPs. It can be seen from Figure 8A that the addition of copper ions had little effect on the morphology of nano-silver. Figure 8B showed TEM images of AgNPs in the presence of Hg^2+^, the addition of Hg^2+^ induced the increase in size and the aggregation of nanoparticles, The diameters of the AgNPs were in the range of 50–200 nm and the average particle size was 150 nm, as shown in Figure 8C. EDS showed three peaks in the elemental analysis in Figure 8D, which was consistent with our analysis.

### 3.5. Sensitivity and Selectivity for S^2−^ Detection

The sensitivity of CMS/AgNPs colloidal solution for S^2−^ was investigated. The absorption intensity of AgNPs decreases gradually along with the addition of S^2−^ (Figure 9A), owing to the formation of Ag_2_S [23,26]. Based on Figure 9A, a favorable linear correlation (R = 0.992) can be obtained with the S^2−^ concentration from 15 to 70 μM (Figure 9B), and the LOD for S^2−^ was estimated to be 2.4 × 10^−7^ M. It can be seen in inset Figure 9A that the color of the sensing system changed from clear yellow to brown with increasing concentration of S^2−^ and, finally, the color changed to colorless when the addition of S^2−^ (70 μM), which provides a platform for a colorimetric method for S^2−^ detection by the naked eyes.

To evaluate the selectivity of this sensing system, 15 kinds various negative ions, such as NO_3_^−^, NO_2_^−^, CO_3_^2−^, HCO_3_^−^, HPO_4_^2−^, H_2_PO_4_^−^, PO_4_^3−^, F^−^, Cl^−^, Br^−^, I^−^, SO_3_^2−^, S_2_O_8_^2−^, S^2−^ and SO_4_^2−^ ions (50 μM), were added to CMS/AgNPs colloidal solution, respectively. Figure 10B gave the absorption spectra changes of AgNPs after the addition of negative ions. Followed by the addition of S^2−^, the absorption peak of AgNPs decreased significantly, the ΔA was 0.65 in Figure 10C, with the color changing from yellow to brown (Figure 10A), while there were no obvious changes for other negative ions. In addition, the relative absorbance values were analogous to that of the S^2−^ value in the presence of all anions. These results clearly indicated that other related anions did not interfere with the spectral and colorimetric detection of S^2−^. These results demonstrate that our proposal shows distinguishing selectivity toward S^2−^.

It can be seen from Figure 11A that the addition of SO_4_^2−^ had little effect on the morphology of nano-silver. From Figure 11B, we can see that the morphology of the nano-silver particles had changed from the original sphere to the amorphous and the particles became larger in the presence of S^2−^. The diameters of the AgNPs were in the range of 40–80 nm and the average particle size was about 60 nm, as shown in Figure 11C. EDS showed a peak for S was observed in the elemental analysis in Figure 11D, which further indicated the presence of silver sulfide.

### 3.6. Detection of Hg^2+^ and S^2−^ in Tap Water and Lake Water

To test the feasibility and practicability of this method, the prepared product was applied to monitor the tap water and lake water. The tap water and lake water samples were taken from our lab and Jian Lake of our university, and then used directly without any pretreatment. The tested water samples (150 µL) were added to CMS/AgNPs colloidal solution (3 mL) with constant stirring, and then were measured with UV–VIS spectrophotometer. However, we have not found any color changes and absorption spectra, indicating that there are no Hg^2+^ and S^2−^ in the tested water samples. Then, we measured the recoveries of Hg^2+^ and S^2−^ in real samples, and the analytical result is in line with our acceptable range in 95.6%–103.8% (Table 1). The performance of our synthesized sensor identified for selective Hg^2+^ and S^2−^ detection was also compared with some previously reported results (Appendix A). This result indicates that our proposed strategy offers the promise for broader practical application in detecting Hg^2+^ and S^2−^, suggesting that the academic study could eventually drive real application and benefit for our environment.

## 4. Conclusion

In summary, we have demonstrated a CMS/AgNPs colloidal colorimetric sensor for highly sensitive and selectivity toward Hg^2+^ and S^2−^ over other positive/negative ions, which showed good linear relationships for Hg^2+^ (0–50 μM) and S^2−^ (15–75 μM), with a low LOD of 1.8 × 10^−8^ M and 2.4 × 10^−7^ M, respectively. Impressively, compared with other tactics, this proposed strategy displayed general fabrication, higher selectivity, wider linear range, lower detection limit, high stability, and environmental friendship. In addition, the fabricated sensor was employed to detect of the target ions of tap and lake water specimens with a satisfactory testing result. Hence, a simple, high selectivity, rapid and efficient strategy could be easily applied for detecting cations (Hg^2+^) and anions (S^2−^) and open up new opportunities for selecting other kinds of ions.

## Figures and Tables

**Figure 1 polymers-12-00113-f001:**
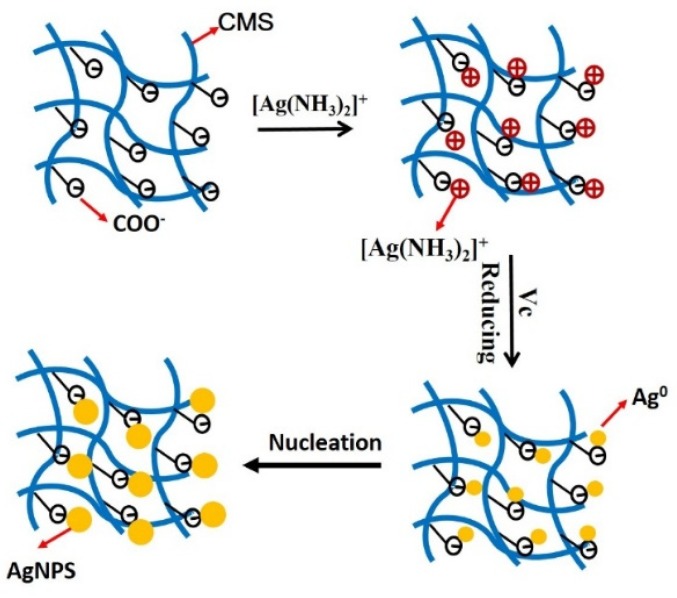
Reaction mechanisms for CMS/AgNPs synthesis.

**Figure 2 polymers-12-00113-f002:**
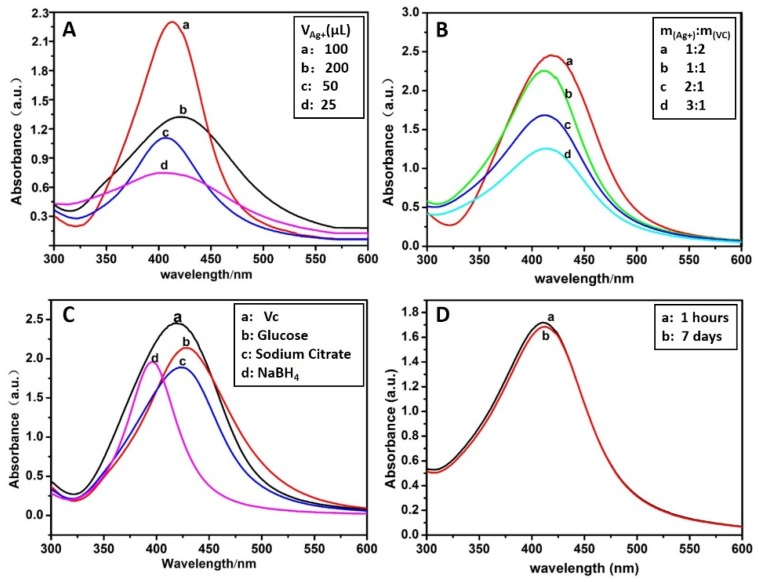
UV–VIS spectra of CMS/AgNPs: (**A**) Different volumes of Tollens’ reagent (0.1 M), (**B**) different proportion of VC, (**C**) different reducing agents, and (**D**) reduction stability of Vc.

**Figure 3 polymers-12-00113-f003:**
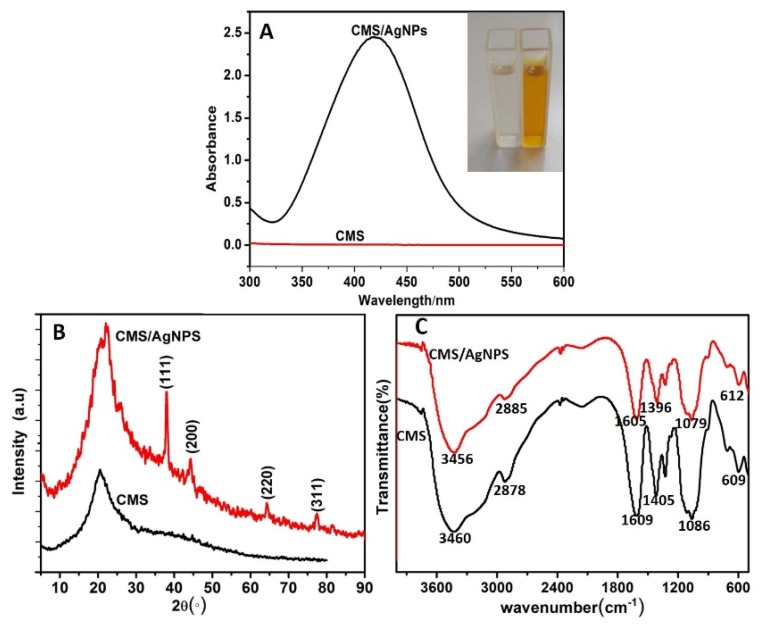
(**A**) UV–VIS spectroscopy, (**B**) XRD pattern, and (**C**) FT-IR spectra of CMS/AgNPs.

**Figure 4 polymers-12-00113-f004:**
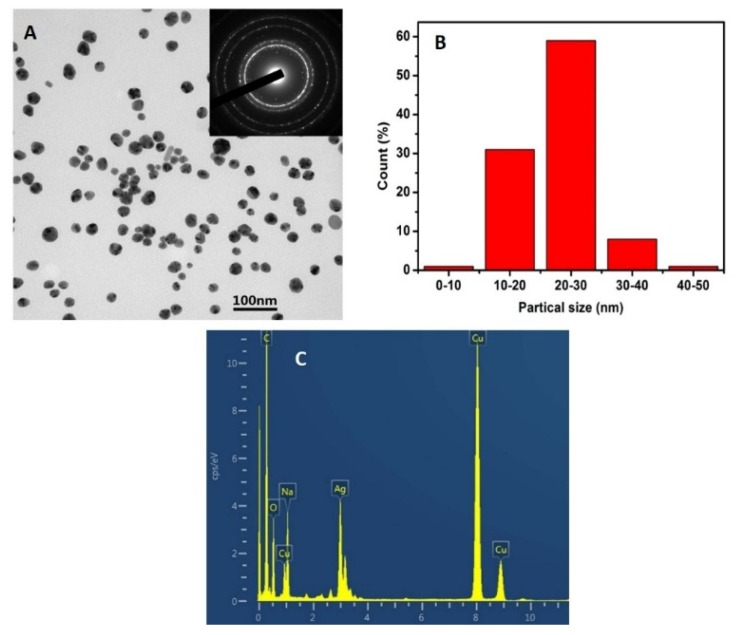
(**A**) TEM images (inset: SAED pattern), (**B**) size distribution, and (**C**) EDS spectrum of CMS/AgNPs.

**Figure 5 polymers-12-00113-f005:**
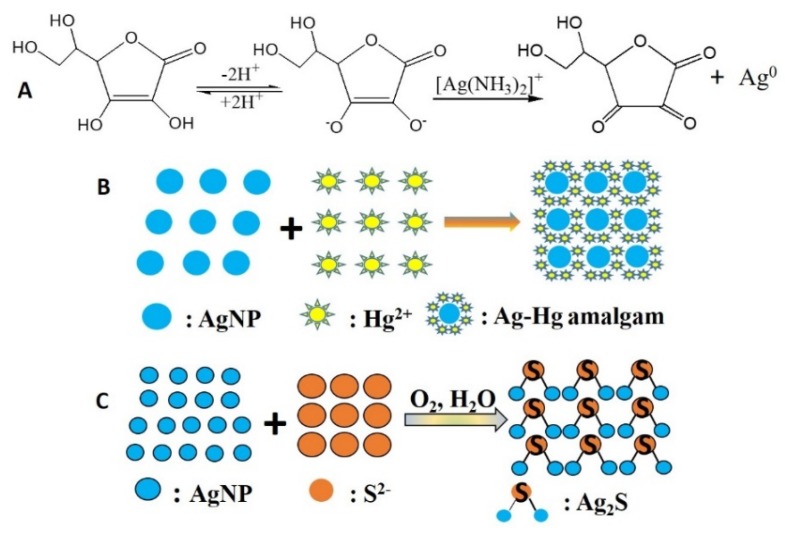
Reaction mechanisms for Vc and Tollens’ reagent (**A**), reaction scheme of AgNPS and Hg^2+^ ions (**B**), and AgNPS and S^2−^ ions (**C**).

**Figure 6 polymers-12-00113-f006:**
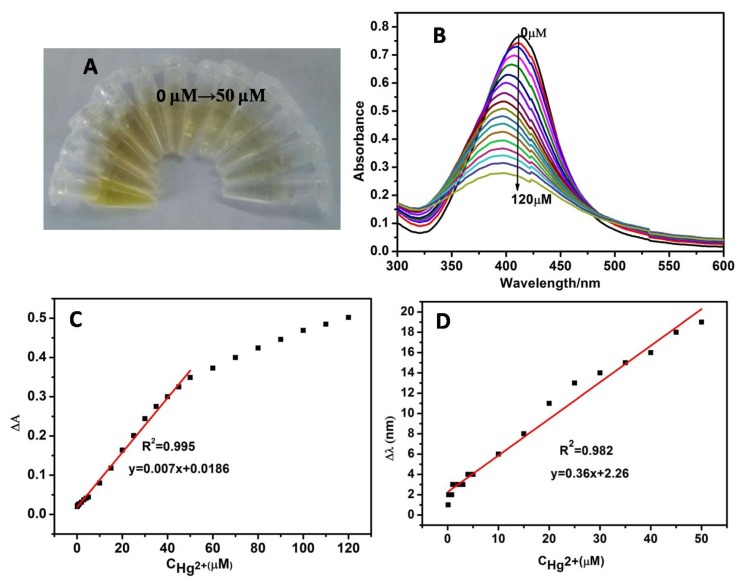
(**A**) The picture of color change and (**B**) UV–VIS spectroscopy of CMS/AgNPs colloidal solution with different concentration of Hg^2+^, (**C**) the liner relative absorption intensity ΔA, and (**D**) the SPR band shift Δl over the concentration of Hg^2+^ is within the range of 0–50 μM.

**Figure 7 polymers-12-00113-f007:**
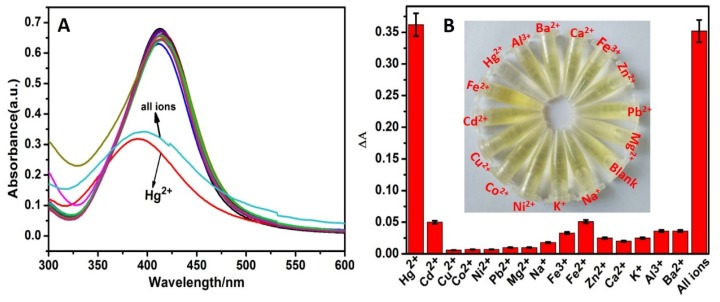
(**A**) UV–VIS absorption spectra and, (**B**) ΔA of the proposed sensing system in the presence of 15 kinds various metal ions and mixed all ions (the concentrations of all metal ions are 50 μM).

**Figure 8 polymers-12-00113-f008:**
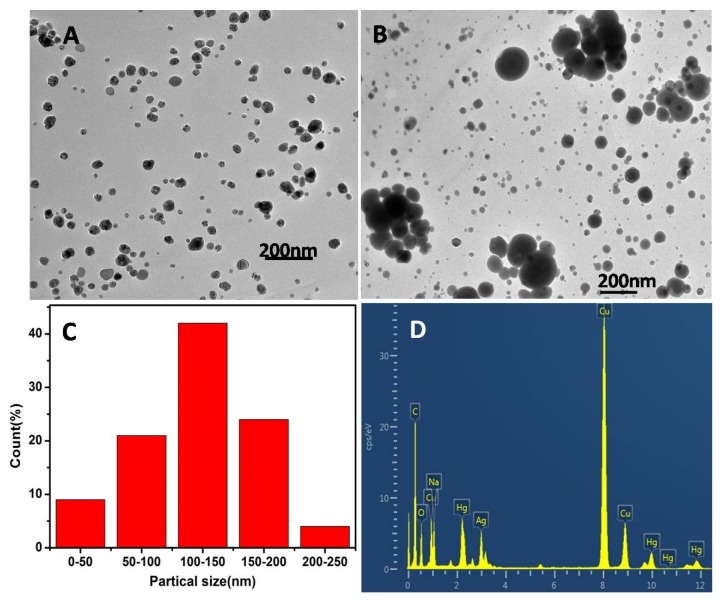
(**A**) TEM of CMS/AgNPs colloidal solution in the presence of Cu^2+^ and (**B**) Hg^2+^ (30 μM), (**C**) size distribution, and (**D**) EDS spectrum of CMS/AgNPs with addition of Hg^2+^.

**Figure 9 polymers-12-00113-f009:**
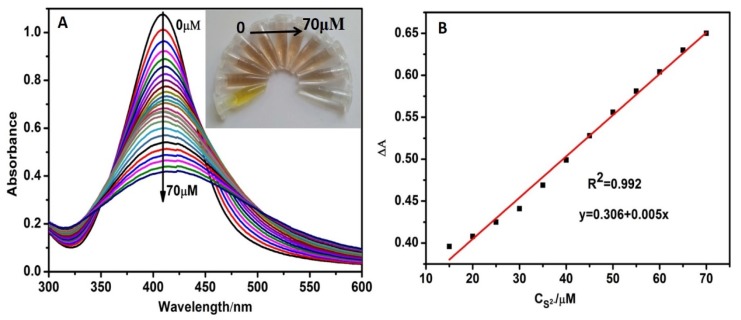
(**A**) UV–VIS spectroscopy of CMS/AgNPs colloidal solution with different concentration of S^2−^, and (**B**) the linear relative absorption intensity ΔA over the concentration of S^2−^ is within the range of 15–70 μM.

**Figure 10 polymers-12-00113-f010:**
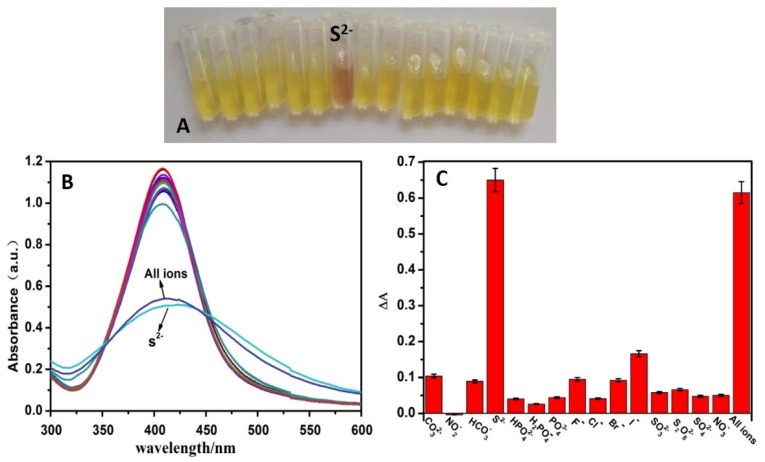
(**A**) The color changes picture, (**B**) UV–VIS absorption spectra, and (**C**) ΔA of the proposed sensing system in the presence of 15 kinds various negative ions and mixed all ions (the concentrations of all negative ions are 50 μM).

**Figure 11 polymers-12-00113-f011:**
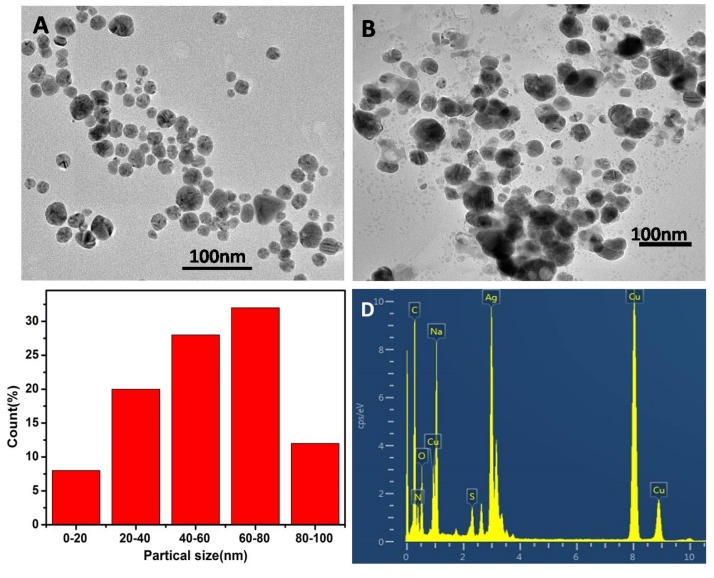
(**A**) TEM of CMS/AgNPs colloidal solution in the presence of SO_4_^2−^ and (**B**) S^2−^ of 50 μM, (**C**) size distribution, and (**D**) EDS spectrum of CMS/AgNPs with addition of S^2−^.

**Table 1 polymers-12-00113-t001:** Analytical results of Hg^2+^ and S^2−^ in tap water and lake water samples (n = 3).

Sample	Hg^2+^ Added	(μM) Found	Recovery (%)	S^2−^ Added	(μM) Found	Recovery (%)
Tap water	0	0	-	0	0	-
	5.00	5.06	101.20	5.00	4.87	97.40
	10.00	9.86	98.60	10.00	10.34	103.40
Lake water	0	0	-	0	0	-
	5.00	4.78	95.60	5.00	5.19	103.80
	10.00	9.61	96.10	10.00	9.83	98.30

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
