# Peer review of "A Green, Rapid and Efficient Dual-Sensors for Highly Selective and Sensitive Detection of Cation (Hg2+) and Anion (S2−) Ions Based on CMS/AgNPs Composites"

_polymers, 2020, doi:10.3390/polym12010113_

Round 1

Reviewer 1 Report

I quite like the idea of this manuscript, which is to detect mercury cations and sulfide anions using carboxymethyl cellulose sodium/silver nanoparticles. The authors have identified a high priority problem, the dual detection of anions and cations in complex environments, and an innovative solution to solve that problem. Nonetheless, there are a number of issues that in my opinion need to be addressed prior to publication of this manuscript. These include:

In the abstract, the authors use the term ‘biosensor.’ This is not defined, nor is it clear why the authors need a ‘biosensor’ for their stated goal rather than simply a high-performance sensor. This should be qualified and explained further. In all cases, the authors should be encouraged to use superscripts to denote the charges on the ions rather than including the charges at full size. The current full-size hampers readability of this manuscript. In general, this manuscript would benefit from a thorough proofreading by a native English speaker. While it is certainly understandable in its current state, there are some syntax-based errors that should be corrected. In the introduction, the authors use a colloquial term “as well all known.” The term is “as we all know,” rather than “as we all known,” and neither term should be used in the formal research paper context. A more concerning problem is the fact that the authors indicate that methods for Hg2+ and S2- detection all require expensive instrumentation, and they ignore the fact that simpler methods for detection of these ions have already been reported. For example, the authors should familiarize themselves with papers such as:

Guo, Lulu; Song, Yonghai; Cai, Keying; Wang, Li "On-​off" ratiometric fluorescent detection of Hg2+ based on N-​doped carbon dots-​rhodamine B@TAPT-​DHTA-​COF. Spectrochim. Acta A 2019, Ahead of Print; DOI: 10.1016/j.saa.2019.117703

Zhang, Yao; Zhang, Lan; Wang, Luyang; Wang, Guoqing; Komiyama, Makoto; Liang, Xingguo Colorimetric determination of mercury(II) ion based on DNA-​assisted amalgamation: a comparison study on gold, silver and Ag@Au Nanoplates. Microchim. Acta 2019, 186, 1-8.

Shojaeifard, Zahra; Hemmateenejad, Bahram; Shamsipur, Mojtaba; Ahmadi, Raheleh Dual fluorometric and colorimetric sensor based on quenching effect of copper (II) sulfate on the copper nanocluster for determination of sulfide ion in water samples. J. Photochem. Photobiol. A: Chem. 2019, 384, 112030. 

Engel, Laura; Tarantik, Karina R.; Pannek, Carolin; Woellenstein, Juergen Screen-​printed sensors for colorimetric detection of hydrogen sulfide in ambient air. Sensors 2019, 19, 1182. 

And many others like this. They should explain more clearly how their work fits into the context of work that has already been published in this area.

The authors refer to an environmentally benign reducing agent, “Vc.” This agent needs to be more clearly delineated the first time it is used. The colloidal solution depends on electrostatic attraction between the carboxylic acid groups on the CMS and the Ag nanoparticles. If so, it seems that it is likely to have a strong pH dependence as well. Has this dependence been analyzed/ is the pH carefully controlled? More details should be provided to answer this question. More information about the non-zero signals for cadmium (2+) and iron (2+) should be provided (see Figure 5B), especially around an explanation for this non-ideal selectivity and how such interfering ions are dealt with in complex environments. The authors are using a mixture of silver nanoparticles and carboxymethylcellulose, but the role of each in the observed performance is not clearly elucidated. The authors need to have control experiments that use just the carboxymethylcellulose and just the silver nanoparticles, and explain from there how the beneficial effect of both components of the sensor can occur. Similarly, the size of the silver nanoparticles appears to be somewhat arbitrarily chosen. More information about how such a size was selected and how the choice of particle size influences detection performance should be provided as well. Finally, there are different ways of connecting silver nanoparticles to carboxymethylcellulose. The authors are using non-covalent electrostatic interactions. It would benefit this paper to investigate other mechanisms of attachment and how the detection performance changes based on the differences in the attachment mechanisms provided.

Overall, while the paper is interesting and the idea is good, there are some minor issues that need to be addressed. Moreover, there are some more major issues around the lack of control experiments that also reduce my overall enthusiasm for this manuscript as currently written.

Author Response

Manuscript No. polymers-649632

Title: "A green, rapid and efficient dual-sensors for highly selective and sensitive detection of cation (Hg2+) and anion (S2-) ions based on CMS/AgNPs composites"

Dear professors,

Many thanks for your greatly helpful comments and suggestions on the manuscript. They will increase the level of the article greatly. We have modified the manuscript accordingly, and detailed corrections are listed in the manuscript.

Reviewer #1:

I quite like the idea of this manuscript, which is to detect mercury cations and sulfide anions using carboxymethyl cellulose sodium/silver nanoparticles. The authors have identified a high priority problem, the dual detection of anions and cations in complex environments, and an innovative solution to solve that problem. Nonetheless, there are a number of issues that in my opinion need to be addressed prior to publication of this manuscript. These include:

In the abstract, the authors use the term ‘biosensor.’ This is not defined, nor is it clear why the authors need a ‘biosensor’ for their stated goal rather than simply a high-performance sensor. This should be qualified and explained further.

Answer:Thanking for giving the significant comment to our research work. This is a constructive comment and we have revised the manuscript accordingly.

The sentence “How to design and fabricate the high-performance biosensor for simultaneously and accurately detecting the Hg2+ and S2− is critical.” has been revised into “How to design and fabricate the high-performance sensor for simultaneously and accurately detecting the Hg2+ and S2− is critical.”

In all cases, the authors should be encouraged to use superscripts to denote the charges on the ions rather than including the charges at full size. The current full-size hampers readability of this manuscript.

Answer:Thanking for giving the precious comment to our research work. This is a constructive comment, and we have revised the manuscript accordingly.

In general, this manuscript would benefit from a thorough proofreading by a native English speaker. While it is certainly understandable in its current state, there are some syntax-based errors that should be corrected. In the introduction, the authors use a colloquial term “as well all known.” The term is “as we all know,” rather than “as we all known,” and neither term should be used in the formal research paper context.

Answer:Thanking for giving the precious comment to our research work. This is a meaningful suggestion, and we have modified it in the paper, meanwhile, we have carefully examined throughout the paper and revised accordingly.

A more concerning problem is the fact that the authors indicate that methods for Hg2+ and S2- detection all require expensive instrumentation, and they ignore the fact that simpler methods for detection of these ions have already been reported. For example, the authors should familiarize themselves with papers such as:

Guo, Lulu; Song, Yonghai; Cai, Keying; Wang, Li "On-off" ratiometric fluorescent detection of Hg2+ based on N-doped carbon dots-rhodamine B@TAPT-DHTA-COF. Spectrochim. Acta A 2019, Ahead of Print; DOI: 10.1016/j.saa.2019.117703

Zhang, Yao; Zhang, Lan; Wang, Luyang; Wang, Guoqing; Komiyama, Makoto; Liang, Xingguo Colorimetric determination of mercury(II) ion based on DNA-​assisted amalgamation: a comparison study on gold, silver and Ag@Au Nanoplates. Microchim. Acta 2019, 186, 1-8.

Shojaeifard, Zahra; Hemmateenejad, Bahram; Shamsipur, Mojtaba; Ahmadi, Raheleh Dual fluorometric and colorimetric sensor based on quenching effect of copper (II) sulfate on the copper nanocluster for determination of sulfide ion in water samples. J. Photochem. Photobiol. A: Chem. 2019, 384, 112030. 

Engel, Laura; Tarantik, Karina R.; Pannek, Carolin; Woellenstein, Juergen Screen-​printed sensors for colorimetric detection of hydrogen sulfide in ambient air. Sensors 2019, 19, 1182. 

And many others like this. They should explain more clearly how their work fits into the context of work that has already been published in this area.

Answer:Thanking for giving the precious comment to our research work. This is a meaningful suggestion, and we have summarized these methods in the paper, as shown in Table S2.

Table S2 Comparison of the proposed Hg2+/S2- detection method with other reported methods

Methods

Probe

Ions

Linear range/(µM)

LOD/nm

Ref.

Colorimetric

β-CD AgNPs

Hg2+

S2-

2.6-250

0.03-0.3

37.5

0.9

[1]

Colorimetric

SSA-AgNPs

Hg2+

S2-

0-5

0-0.8

14

4

[2]

Colorimetric

AUNSs

Hg2+

0.1-100

0.05

[3]

Electro chemistry

AgNPs

Hg2+

5.0-755

0.19

[4]

Colorimetric

AgNPs

Hg2+

0.5-5.0

58

[5]

Colorimetric

SA-AgNPs

Hg2+

0.025-60

5.29

[6]

Fluorescence

NCDs-RhB@COF

Hg2+

0.048-10

15.9

[7]

Electro chemistry

SPCE/Go/AuNPs

S2-

3-40

300

[8]

Colorimetric

Cip-AgNPs

S2-

-

16

[9]

Colorimetric

AgNPs

S2-

3-40

300

[10]

Colorimetric

CMS/AgNPs

Hg2+

S2-

0-50

15-70

18

240

This work

Rajamanikandan, R. and M. Ilanchelian, β-cyclodextrin functionalised silver nanoparticles as a duel colorimetric probe for ultrasensitive detection of Hg2+ and S2− ions in environmental water samples. Materials Today Communications, 2018. 15: p. 61-69. Das, S., et al., Sensitive and robust colorimetric assay of Hg2+ and S2− in aqueous solution directed by 5-sulfosalicylic acid-stabilized silver nanoparticles for wide range application in real samples. Journal of Environmental Chemical Engineering, 2017. 5(6): p. 5645-5654. Chen, J.L., et al., Determination of mercury (II) ions based on silver-nanoparticles-assisted growth of gold nanostructures: UV-Vis and surface enhanced Raman scattering approaches. Spectrochim Acta A Mol Biomol Spectrosc, 2018. 199: p. 301-307. Eksin, E., et al., Ecofriendly Sensors Developed by Herbal Based Silver Nanoparticles for Electrochemical Detection of Mercury (II) Ion. Electroanalysis, 2019. 31(6): p. 1075-1082. Ertürk, A.S., Biosynthesis of Silver Nanoparticles Using Epilobium parviflorum Green Tea Extract: Analytical Applications to Colorimetric Detection of Hg2+ Ions and Reduction of Hazardous Organic Dyes. Journal of Cluster Science, 2019. 30(5): p. 1363-1373. Faghiri, F. and F. Ghorbani, Colorimetric and naked eye detection of trace Hg(2+) ions in the environmental water samples based on plasmonic response of sodium alginate impregnated by silver nanoparticles. J Hazard Mater, 2019. 374: p. 329-340. Guo, L., et al., "On-off" ratiometric fluorescent detection of Hg(2+) based on N-doped carbon dots-rhodamine B@TAPT-DHTA-COF. Spectrochim Acta A Mol Biomol Spectrosc, 2019: p. 117703. Chen, Y.-H., et al., Electrodeposited Ag, Au, and AuAg nanoparticles on graphene oxide-modified screen-printed carbon electrodes for the voltammetric determination of free sulfide in alkaline solutions. Electrochimica Acta, 2016. 205: p. 124-131. Ayaz Ahmed, K.B., M. Mariappan, and A. Veerappan, Nanosilver cotton swabs for highly sensitive and selective colorimetric detection of sulfide ions at nanomolar level. Sensors and Actuators B: Chemical, 2017. 244: p. 831-836. Zhao, L., et al., A Colorimetric Sensor for the Highly Selective Detection of Sulfide and 1,4-Dithiothreitol Based on the In Situ Formation of Silver Nanoparticles Using Dopamine. Sensors, 2017. 17(3): p. 626.

The authors refer to an environmentally benign reducing agent, “Vc.” This agent needs to be more clearly delineated the first time it is used.

Answer:Thanking for giving the precious comment to our research work. This is a meaningful suggestion, and we have added some description in the paper.

“Vc is a water-soluble vitamin and non-toxic reducing agent. There are double O-H with C=C bonds in the structure of Vc, so it has strong reducing ability.”

The colloidal solution depends on electrostatic attraction between the carboxylic acid groups on the CMS and the Ag nanoparticles. If so, it seems that it is likely to have a strong pH dependence as well. Has this dependence been analyzed/ is the pH carefully controlled?

Answer:Thanking for giving the precious comment to our research work. This is a meaningful suggestion, and we have investigated the influence of the pH, the detail description can be found in the manuscript and supplementary materials.

Fig. S2 (A) UV–vis spectra of CMS/AgNPs and (B) ΔA under different PH

Furthermore, the UV–vis spectral responses of CMS/AgNPs were recorded at different pH in Fig. S1. The pH of synthesized CMS/AgNPs sol is 8.39. The CMS/AgNPs shows no noticeable effects with the change in pH from 6.59 to 10.51. However, the CMS/AgNPs shows spectral changes at low pH (< 2.48) due to the COO- in the CMS combined with H+ to form COOH, which reduced the stabilizing of AgNPs. These results confirm that CMS can stabilize AgNPs because of electrostatic adsorption between COO- and [Ag (NH3)2]+.

More details should be provided to answer this question. More information about the non-zero signals for cadmium (2+) and iron (2+) should be provided (see Figure 5B), especially around an explanation for this non-ideal selectivity and how such interfering ions are dealt with in complex environments. The authors are using a mixture of silver nanoparticles and carboxymethylcellulose, but the role of each in the observed performance is not clearly elucidated. The authors need to have control experiments that use just the carboxymethylcellulose and just the silver nanoparticles, and explain from there how the beneficial effect of both components of the sensor can occur. Similarly, the size of the silver nanoparticles appears to be somewhat arbitrarily chosen.

Answer:Thanking for giving the precious comment to our research work. This is a meaningful suggestion, and we have added some description in the paper.

Fig. 7 (A) UV–vis absorption spectra and (B) ΔA of the proposed sensing system in the presence of 15 kinds various metal ions and mixed all ions. (The concentrations of all metal ions are 50 μM)

To evaluate the selectivity of CMS/AgNPs colloidal solution, besides Hg2+, other 14 kinds of metal ions, including Cd2+, Co2+, Cu2+, Fe3+, Fe2+, K+, Al3+, Ba2+, Ca2+, Mg2+, Na+, Ni2+, Pb2+ and Zn2+ were selected. The absorption spectra of AgNPs with various metal ions were given in Fig. 7A. Comparing these plots, only the Hg2+ one had response for AgNPs, and upon the addition of Hg2+, the absorption peak decreased and exhibited an obvious blue shift. While for other 14 metal ions, the absorption spectrums were almost unchanged even at the maximum concentration (50 μM). As shown in Fig. 7B, the change of ΔA was 0.36 in the presence of Hg2+, but there was no obvious change observed in the presence of other metal ions. The color of the sensing system did not change after the addition of other metal ions (50 μM) except for Hg2+, which made the system from yellow to colorless (inset, Fig. 7B). In addition, in the presence of all anions, the relative absorbance values were analogous to that of Hg2+ value. These results clearly indicated that other related anions did not interfere with the spectral and colorimetric detection of Hg2+. The experimental results verify that our proposal shows unique selectivity for Hg2+.

Fig. 10 (A) The color changes picture, (B) UV–vis absorption spectra and (C) ΔA of the proposed sensing system in the presence of 15 kinds various negative ions and mixed all ions. (The concentrations of all negative ions are 50 μM).

To evaluate the selectivity of this sensing system, 15 kinds various negative ions such as NO3-, NO2-, CO32-, HCO3-, HPO42-, H2PO4-, PO43-, F-, Cl-, Br-, I-, SO32-, S2O82-, S2- and SO42- ions (50 μM) were added to CMS/AgNPs colloidal solution respectively. Fig. 10B gave the absorption spectra changes of AgNPs after addition of negative ions. Followed by the addition of S2-, the absorption peak of AgNPs decreased significantly, the ΔA was 0.65 in Fig. 10C with the color changes from yellow to brown (Fig. 10A), while there were no obvious changes for other negative ions. In addition, the relative absorbance values were analogous to that of S2- value in the presence of all anions. These results clearly indicated that other related anions did not interfere with the spectral and colorimetric detection of S2-. These results demonstrate that our proposal shows distinguishing selectivity toward S2-.

In this paper, carboxymethyl cellulose sodium/silver nanoparticles (CMS/AgNPs) colloidal solution show highly detection for Hg2+ and S2− due to the synergy effect of the two compounds, while the CMS was utilized as both the stabilizer and the hydrophilic substrate for AgNPs. Just the CMS without AgNPs did not show any signals for the detection of metal ions, and the AgNPs without CMS are easy to aggregate into large particles due to the high surface-energy, thus affecting the detective results.

We have conducted a series of parallel experiments, the same experiment condition with the certain reductant can achieve a uniform size of the silver nanoparticles for 20 nm.

More information about how such a size was selected and how the choice of particle size influences detection performance should be provided as well.

Answer:Thanking for giving the precious comment to our research work. This is a meaningful suggestion, and we have added some description in the paper.

Fig. S2 TEM images of different reducing agent: (A) NaBH4, (B) Sodium Citrate and (C) Glucose

Table S1 Comparison of the proposed Hg2+ detection method with different reducing agent

Reducing agent

Size of AgNPS/(nm)

Linear range/(µM)

LOD/(µM)

NaBH4

~10

20-120

0.32

Glucose

~40

0-60

0.12

Sodium Citrate

~30

0-50

0.013

Vc

~20

0-50

0.018

From Fig. S2, we could see that the stronger the reducing ability of the reducing agent, the smaller the average particle size of the synthesized AgNPs. Table S1 showed that the smaller the particle size of AgNPs, the lower the LOD of Hg2+. The reason is that the particles are small, and the stability is not good, so low concentration of Hg2+ will cause aggregation of AgNPs.

Finally, there are different ways of connecting silver nanoparticles to carboxymethylcellulose. The authors are using non-covalent electrostatic interactions. It would benefit this paper to investigate other mechanisms of attachment and how the detection performance changes based on the differences in the attachment mechanisms provided.

Answer:Thanking for giving the precious comment to our research work. This is a constructive comment, which has pointed the direction for the further investigation.

Overall, while the paper is interesting and the idea is good, there are some minor issues that need to be addressed. Moreover, there are some more major issues around the lack of control experiments that also reduce my overall enthusiasm for this manuscript as currently written.

Answer:Thanking for giving the precious comment to our research work. We have conducted a series of control experiments, such as the proportion of reactants, the choice of reductant or some related measurements, the detail analysis is shown in the manuscript.

Fig. 2 UV–vis spectra of CMS/AgNPs: (A) Different volumes of Tollens’ reagent (0.1M), (B) Different proportion of VC (C) Different reducing agents and (D) reduction stability of Vc

Fig. 2 is the optimal screening of the experiment. The absorbance of AgNPs increases with the increase of the amount of silver ions in Fig. 2A. The absorbance of AgNPs decreases and red shift occurs in 200 μL Tollens’ reagent, indicating that the nano-silver particles become larger at a higher concentration, so the best amount is 100 μL Tollens’ reagent. The absorbance of AgNPs increases with the increase of the amount of Vc in Fig. 2B. The absorbance of AgNPs decreases and red shift occurs in 200 μL Tollens’ reagent, indicating that the nano-silver particles become larger at a higher concentration, so the best amount is 100 μL Tollens’ reagent. Fig. 2C is UV–vis spectra of CMS/AgNPs by synthesized with different reducing agents, indicating that Vc is weaker in reducing ability than NaBH4, but stronger than glucose and sodium citrate. The latter two require a catalyst for the reaction to synthesize AgNPs. The synthesized AgNPs with NaBH4 was weaker in stability than Vc, so Vc was the best and green reducing agents. The stability of the synthesized AgNPs is optimal in Fig. 2D. We can see from Fig. 2 that the best experimental conditions are CMS aqueous solution (0.1% [w/v], 100 mL) and 1:1 molar ratio of Tollens’ reagent (0.1 M, 100 μL) to Vc (0.1 M, 100 μL). Further, the experiments show that Vc is weaker in reducing ability than B, but stronger than glucose and sodium citrate. The latter two require a catalyst for the reaction.

Fig. 3C shows the FT-IR spectra of CMS and CMS/AgNPs. For CMS/AgNPs, the band centered at around 3456 cm-1 can be attributed to the stretching vibration of hydroxyl group; the band at about 2885 cm-1 is assigned to the C–H group; the band at around 1605 cm-1 is attributed to the stretching vibration of carboxyl group; the band at around 1396 cm-1 is corresponded to the C–H bending mode; the absorption band at 1079 cm-1 is ascribed to C–O–C stretching mode from the glucosidic units; the peak at 612cm-1 was related to the deformation vibration of hydrogen bond. Fig. 3C shows the FT-IR spectra of CMS which is similar to the CMS/AgNPs, indicating that there is no chemical reaction between the formed AgNPs and CMS.

Reviewer 2 Report

The manuscript presents a green efficient dual sensor for detection of Hg2+ and S2- ions based on a CMS/Ag nanoparticles composite. Its sensitivity is clear and Absorbance measurements reveal undoubted selectivity. Green materials and procedures make this sensor environment friendly. The research design is modest but adequate if the paper intends to be just a colorimetric study, although other kinds of analyses (such as IR, XPS, ...) could be useful to complement the obtained results.

A a major fault, a weak comparison with previous work does not make clear the contribution of this paper to the knowledge in the field. The only reference in the Results and discussion section is cited only to support the Ag-Hg amalgam, and no discussion is presented to compare present results with the state-of-the-art. The manuscript fails stating its highlights and convincing of the novelty of this research, as compared to previous reports.

By the way, I add some less important comments:

Superscripts and subscripts are lost along the whole paper (except in the Title), making it more difficult to see charges in ions and exponents in scientific notation (see for instance the Abstract). Please mark correctly superscripts and subscripts.

Also, histrograms for particle sizes visible by TEM in the cases of Hg2+ and S2- should be added to figures 6 and 9, respectively, to support the changes stated in the Mechanisms section (line 224).

Similarly, I think that any reaction scheme could help understanding the detection mechanisms in section 3.5.

Besides, please revise the Engligh language and shorten sentences joined by commas (like the first sentence in section 3.5). 

Finally, please revise the Acknowledgments section, as it keeps a general explanation provided in the template.

Author Response

Manuscript No. polymers-649632

Title: "A green, rapid and efficient dual-sensors for highly selective and sensitive detection of cation (Hg2+) and anion (S2-) ions based on CMS/AgNPs composites"

Dear professors,

Many thanks for your greatly helpful comments and suggestions on the manuscript. They will increase the level of the article greatly. We have modified the manuscript accordingly, and detailed corrections are listed in the manuscript.

Reviewer 3 Report

Dear Author,

I think that your work describes a valid example of real-world oriented design of a chemical sensor, and in this sense I find it valuable and worthy of further investigations. Before considering the publication in this Journal, please address the following aspects.

1) Your draft needs a careful English spell check, e.g., r.82, 148-149, 176.

2) Abstract and Introduction. Please clarify the field of application of your sensor, i.e., environmental monitoring or clinic diagnosis.

3) r.28. Given your detection strategy, what other cations and anions species are you planning to detect? What kind of nanoparticles would you adopt?

4) I strongly suggest that you anticipate Par. 3.5 before Par. 3.3, as this would be beneficial for detection strategy understanding by the readers.

5) rr.154, 159: please substitute “∆l” with “∆λ”.

6) Fig. 4B-C. I assume that you evaluate ∆A as the difference between absorbance peak in presence of target vs. the blank measurement, i.e., 0 μM. If so, from the data of Fig. 4B it seems that the ∆A @ 50 μM is 0.5, while in Fig. 4C it is less than 0.35. Please clarify this point.

7) Fig. 4 and 7. Please add error bars to your calibration curves to prove the repeatability of your sensors. What is the numerosity of your experiments?

8) Please show the “∆λ” calibration curve in presence of S (Fig. 7).

Author Response

(The authors gave the same response as above.)

Round 2

Reviewer 1 Report

The authors have done a good job responding to my comments on the previous version of the manuscript, and I recommend that the manuscript be accepted for publication after the following, relatively minor issues are addressed:

In the introduction, the authors indicate that ‘we all know’ that H2S is harmful to our eyes. This may be true that they are all aware of this fact, but it is informal to write ‘we all know’ in the context of a scientific manuscript. A reference that supports this assertion is more than sufficient to indicate that ‘we all know’ a certain fact. The authors posit that the reason why smaller nanoparticles provide lower LODs is that the “stability is not good” of the particles. More information about these stability limitations should be provided.

Author Response

Dear professors,

Many thanks for your greatly helpful comments and suggestions on the manuscript. They will increase the level of the article greatly. We have modified the manuscript accordingly, and detailed corrections are listed in the manuscript.

Reviewer #1:

The authors have done a good job responding to my comments on the previous version of the manuscript, and I recommend that the manuscript be accepted for publication after the following, relatively minor issues are addressed:

In the introduction, the authors indicate that ‘we all know’ that H2S is harmful to our eyes. This may be true that they are all aware of this fact, but it is informal to write ‘we all know’ in the context of a scientific manuscript. A reference that supports this assertion is more than sufficient to indicate that ‘we all know’ a certain fact.

Answer:Thanking for giving the significant comment to our research work. This is a constructive comment and we have revised the manuscript accordingly.

“S2- not only exists hazard to living system but also can easily convert into H2S in acid medium [9], and H2S is harmful to our eyes, causing conjunctivitis and other related eye diseases [10].”

The authors posit that the reason why smaller nanoparticles provide lower LODs is that the “stability is not good” of the particles. More information about these stability limitations should be provided.

Answer:Thanking for giving the significant comment to our research work. This is a constructive comment and we have revised the manuscript accordingly.

This phenomenon could be attributed to the high specific surface areas and surface energy of smaller size of AgNPs, when Hg2+ ions were added to the CMS/AgNPs colloid solution, the contact area and reactivity between AgNP and Hg2+ increases along with the size decrease of AgNPs, thus quickening the adsorption and reaction speed. Even at a low concentration of Hg2+, obvious absorbance signals can be obtained in the CMS/AgNPs colloid solution. These results indicated that the small nanoparticles have good sensitivity and lower LOD.

Reviewer 2 Report

Authors have greatly improved their manuscript, now showing more clearly its value. I think it can be published as is.

Author Response

Dear professors,

Many thanks for your greatly helpful comments and suggestions on the manuscript. They will increase the level of the article greatly.

Best wishes!